# Targeting Glycolysis with 2-Deoxy-D-Glucose and Lysosomal Integrity with L-Leucyl-L-Leucine Methyl Ester as Antimelanoma Strategy

**DOI:** 10.3390/pharmaceutics17101312

**Published:** 2025-10-09

**Authors:** Milica Kosic, Mihajlo Bosnjak, Milos Mandic, Ljubica Vucicevic, Maja Misirkic Marjanovic, Sofie Espersen Poulsen, Ljubica Harhaji-Trajkovic

**Affiliations:** 1Institute of Microbiology and Immunology, Faculty of Medicine, University of Belgrade, Dr. Subotica 1, 11000 Belgrade, Serbia; milica.kosic@med.bg.ac.rs (M.K.); mihajlo.bosnjak@med.bg.ac.rs (M.B.); milos.mandic@med.bg.ac.rs (M.M.); 2Institute for Biological Research “Sinisa Stankovic”-National Institute of Republic of Serbia, University of Belgrade, Bulevar Despota Stefana 142, 11000 Belgrade, Serbia; ljubica.vucicevic@ibiss.bg.ac.rs (L.V.); maja.misirkic@ibiss.bg.ac.rs (M.M.M.); 3Department of Immunology and Microbiology, Faculty of Health and Medical Sciences, University of Copenhagen, Blegdamsvej 3B, 2200 Copenhagen, Denmark; sep@sund.ku.dk

**Keywords:** melanoma, lysosome membrane permeabilization, glycolysis, 2-deoxy-D-glucose, L-leucyl-L-leucine methyl ester, mefloquine, siramesine, cathepsins, mitochondrial dysfunction, oxidative phosphorylation

## Abstract

**Background/Objectives**: Melanoma cells enhance glycolysis and expand lysosomes to support energy metabolism, proliferation, and metastasis. However, lysosomal membrane permeabilization (LMP) causes cathepsin leakage into cytosol triggering cytotoxicity. This study investigated the antimelanoma effect of 2-deoxy-D-glucose (2DG), an inhibitor of glycolytic enzyme hexokinase-2, in combination with cathepsin C-dependent LMP inducer L-leucyl-L-leucine methyl ester (LLOMe) and cathepsin C-independent LMP-inducers mefloquine and siramesine. **Methods**: The viability of A375 and B16 melanoma cells and primary fibroblasts was measured by crystal violet. Apoptosis, necrosis, and LMP were assessed by flow cytometry; caspase activation, mitochondrial depolarization, superoxide production, and energy metabolism were analyzed by fluorimetry, and expression of cathepsins and hexokinase-2 was evaluated by immunoblot. Appropriate inhibitors, antioxidant, and energy boosters were used to confirm cell death type and mechanism. **Results**: LLOMe triggered LMP, mitochondrial depolarization, and mitochondrial superoxide production, while suppressing oxidative phosphorylation. 2DG suppressed glycolysis and, together with LLOMe, synergized in ATP depletion, caspase activation, and mixed apoptosis and necrosis in A375 cells. Inhibitors of lysosomal acidification, cysteine cathepsins, and caspases, as well as antioxidant and energy boosters, reduced 2DG+LLOMe-induced toxicity. Cathepsins B, C, and D were lower, while hexokinase-2 was higher in A375 cells than fibroblasts. Accordingly, 2DG exhibited lower while LLOMe exhibited higher toxicity against fibroblasts than A375 and B16 cells. However, mefloquine and siramesine induced stronger LMP in A375 cells than in fibroblasts and showed melanoma-selective toxicity when combined with 2DG. **Conclusions**: 2DG-mediated glycolysis inhibition in combination with lysosomal destabilization induced by mefloquine and siramesine, but not with non-selectively toxic LLOMe, may be promising antimelanoma strategy.

## 1. Introduction

Melanoma is a highly aggressive skin cancer whose incidence has increased more rapidly than that of any other cancer type since the mid-1950s [1]. Global estimates indicate a current incidence of ~3.3 per 100,000 individuals [2]. While early-stage melanoma can often be successfully treated with surgical excision, metastatic melanoma remains a major therapeutic challenge, with a dismal 5-year survival rate of only 23% [3]. Approximately half of patients with advanced melanoma harbor activating BRAF mutations that drive MEK/ERK pathway activation and initially respond to BRAF inhibitors, but most relapse within months due to acquired resistance [4]. Therefore, there is an urgent need for novel and durable therapeutic strategies.

A hallmark of many cancers, including BRAF mutant melanoma, is their strong reliance on aerobic glycolysis to meet the energy demands of rapid growth and proliferation [5,6]. In addition to enabling rapid ATP generation, glycolysis supplies key metabolic intermediates required for biomass synthesis, further fueling tumor expansion. This metabolic dependency renders melanoma cells particularly vulnerable to glycolysis inhibition. For instance, 2-deoxy-D-glucose (2DG), a glucose analog that competitively inhibits the glycolytic enzymes hexokinase (HK) and phosphoglucoisomerase (GPI) [7], has been shown to reduce ATP production. It also inhibits protein N-glycosylation, induces endoplasmic reticulum (ER) stress and oxidative stress, arrests the cell cycle, and ultimately triggers apoptosis of tumor cells [7]. 2DG has been shown to enhance the cytotoxic effects of several conventional [8,9] and experimental antimelanoma agents [10,11,12,13]. Moreover, 2DG synergizes with various anticancer agents in in vivo models of melanoma [13] and non-melanoma tumors [14,15], where it reduces tumor vascularization [14] and enhances antitumor immune responses [15], besides its direct antitumor activity. Importantly, 2DG is well tolerated in combination with chemotherapy and radiotherapy in patients with solid tumors [16,17,18].

In addition to metabolic rewiring, melanoma cells exhibit an increased number, size, and enzymatic content of lysosomes, accompanied by structural instability of the lysosomal membrane [19,20,21]. These altered lysosomes promote invasion and metastasis by secreting hydrolytic enzymes, cathepsins, that degrade the extracellular matrix [19,22]. Lysosomes also contribute to drug resistance by sequestering and exporting weakly basic chemotherapeutics [19,22]. Moreover, as final effectors of the autophagy pathway, lysosomes support tumor growth by degrading damaged organelles and recycling their components into biosynthetic precursors and energy-yielding metabolites [22]. In contrast, elevated lysosomal content and cathepsin levels, together with membrane instability, may render melanoma cells particularly susceptible to lysosomal membrane permeabilization (LMP). LMP causes cathepsin release into the cytosol, where they non-specifically cleave intracellular proteins and initiate cell death [20,22]. Consequently, several LMP-inducing agents have been investigated for melanoma therapy [19,23,24,25,26,27,28,29].

L-leucyl-L-leucine methyl ester (LLOMe) is a lysosomotropic dipeptide that accumulates in lysosomes, where it is processed by cathepsin C into membrane-disrupting polymers [30], and the sensitivity of different cells to LLOMe correlates with cathepsin C levels [31]. Thereafter, LLOMe triggers rapid lysosomal swelling, membrane rupture, and efflux of cysteine proteases cathepsins B and L [31,32] or aspartic protease cathepsin D [33], which in turn activate caspase-dependent apoptosis [31,34], necrosis [9,35,36,37,38], or ferroptosis [39]. Importantly, LLOMe exhibits higher toxicity toward melanoma cells than toward primary melanocytes [40]. LLOMe is shown to potentiate the antitumor effect of inhibitors of cyclin-dependent kinases in an in vivo model of breast cancer [41].

The antimalarial drug mefloquine is a cationic amphiphilic compound that accumulates in lysosomes and integrates into the membrane lipid bilayer [42], leading to LMP with subsequent release of cathepsins B and L [43]. Mefloquine has been shown to inhibit mitochondrial respiration, reduce mitochondrial membrane potential (MMP), and deplete ATP levels, while enhancing reactive oxygen species (ROS) generation, ER stress, and apoptosis in cancer cells [44,45]. In in vivo tumor models, mefloquine suppresses angiogenesis [46], inhibits mitophagy, and exacerbates mitochondrial dysfunction, thereby promoting mitochondria-mediated apoptosis [47]. The experimental antidepressant and lysosomal detergent siramesine induces accumulation of membranolytic sphingomyelin and lysoglycerophospholipids in lysosomes [48,49,50]. This triggers LMP, release of cysteine cathepsins including cathepsin B into cytosol, oxidative stress, loss of MMP, and apoptosis [48,51,52]. In vivo, siramesine exhibits anticancer activity both alone and in combination with conventional cytostatics [52,53,54].

Combination therapy is routinely employed in oncology to simultaneously target multiple pathways, thereby increasing therapeutic impact and limiting resistance evolution [55]. Accordingly, our therapeutic strategy focuses on the simultaneous targeting of two specific vulnerabilities of melanoma cells: their reliance on glycolytic metabolism and the structural instability of their lysosomes. We demonstrated that 2DG and LLOMe synergistically induce severe energy depletion and mixed apoptotic–necrotic death of A375 melanoma cells by inhibiting both glycolysis and oxidative phosphorylation (OXPHOS), with 2DG primarily targeting glycolysis, while LLOMe predominantly impairs OXPHOS through LMP-induced cathepsin release and subsequent mitochondrial dysfunction. However, the therapeutic potential of the combined treatment was compromised by the high toxicity of LLOMe toward primary skin fibroblasts, which express higher levels of cathepsin C than A375 cells. We further identified mefloquine and siramesine, which may represent more suitable candidates for synergistic combination with 2DG in future antimelanoma strategies.

## 2. Materials and Methods

### 2.1. Cell Culture

A375 human melanoma cells with a BRAF^V600E^ mutation (a kind gift from Dr. Jelena Grahovac, Institute for Oncology and Radiology of Serbia), BRAF^wt^ B16 mouse melanoma cells (European Collection of Animal Cell Cultures, Salisbury, UK), and Normal Human Dermal Fibroblasts (NHDF; Merck, Darmstadt, Germany) were cultured at 37 °C in a humidified atmosphere with 5% CO_2_ in high-glucose (4.5 g/L) DMEM with L-glutamine (Capricorn Scientific, Ebsdorfergrund, Germany), supplemented with 10% fetal bovine serum (FBS; Gibco, Thermo Fisher Scientific, Waltham, MA, USA), 1 mM sodium pyruvate, and 1% antibiotic/antimycotic solution (both from Capricorn Scientific). After a 24 h resting period, cells were treated with 2DG (Merck) and/or LLOMe (MedChemExpress, Monmouth Junction, NJ, USA) in the presence or absence of the following compounds: necrostatin-1, ferrostatin-1, wortmannin, Q-VD-OPh, bafilomycin A1, MG132, N-acetylcysteine (NAC), BAPTA-AM, sodium succinate dibasic hexahydrate or L-carnitine (all from Merck), and pepstatin A or superoxide dismutase 1 (SOD1) (both from Santa Cruz Biotechnology, Dallas, TX, USA). Alternatively, cells were treated with 2DG in the presence or absence of mefloquine or siramesine (both from Merck), or with LLOMe in the presence of sodium dichloroacetate (DCA; Merck) or shikonin (MedChemExpress). A375 cells were seeded at 1 × 10^4^ cells/well in 96-well plates for the crystal violet test, 2.5 × 10^4^ cells/well in 96-well plates for oxygen consumption rate (OCR) and extracellular acidification rate (ECAR) measurement, 1 × 10^5^ cells/well in 24-well plates for flow cytometric analysis and fluorimetry, and 2 × 10^6^ cells/well in 100 mm Petri dishes for fluorescent microscopy, cell transfection, and immunoblotting. NHDF cells were seeded at 9 × 10^3^ cells/well in 96-well plates for the crystal violet test, 9 × 10^4^ in 24-well plates for flow cytometry, and 1.5 × 10^6^ in 100 mm Petri dishes for immunoblotting. B16 cells were seeded at 1 × 10^4^ cells/well in 96-well plates for the crystal violet test.

### 2.2. Cell Viability

Cell viability was assessed by staining adherent cells with crystal violet (Merck), as previously described [56].

### 2.3. Synergism Assessment

The type of interaction between the two treatments (additive, synergistic, or antagonistic) was evaluated using the equation α = (SF_2DG_ × SF_LLOMe_)/SF_2DG+LLOMe_, where SF_2DG_ and SF_LLOMe_ represent the surviving fractions after treatment with 2DG and LLOMe, respectively, and SF_2DG+LLOMe_ represents the surviving fraction after the combined treatment. Statistical significance of the α values was determined using a one-sample t-test against the theoretical value of 1 in GraphPad Prism 8.0.2 (GraphPad Software, San Diego, CA, USA). α = 1 indicates an additive effect, α > 1 indicates synergism, and α < 1 indicates antagonism.

### 2.4. Apoptosis/Necrosis Analysis

Apoptotic and necrotic cell death were analyzed upon double staining with Annexin V-FITC and 7-AAD (BD Biosciences, San Diego, CA, USA), according to the manufacturer’s protocol. Annexin V binds to phosphatidylserine on apoptotic cells, while 7-AAD labels necrotic cells with compromised membranes, enabling distinction of viable (Annexin^−^/7-AAD^−^), apoptotic (Annexin^+^/7-AAD^−^), and necrotic (Annexin^+^/7-AAD^+^) populations. DNA fragmentation (sub-G0/G1 compartment), a hallmark of apoptotic cell death, was evaluated by cell cycle analysis of propidium iodide-stained cells exactly as previously described [57]. Both analyses were performed on the FACS Aria III flow cytometer, using FACSDiva 6.0 software for acquisition and FlowJo 10.7 software for analysis (BD Biosciences, San Diego, CA, USA).

### 2.5. Lysosomal Staining

Lysosomes were stained with acridine orange (Thermo Fisher Scientific) or LysoTracker Red (MedChemExpress) according to the manufacturers’ instructions. Acridine orange-stained cells were analyzed using an inverted fluorescent microscope (Leica Microsystems DMIL, Wetzlar, Germany), Leica Microsystems DFC320 camera, and Leica Application Suite software (version 2.8.1), where lysosomes appeared as orange/red cytoplasmic vesicles, while nuclei and cytoplasm were stained green. Alternatively, acridine orange- and LysoTracker Red-stained cells were analyzed by flow cytometry using a FACS Aria III flow cytometer. For LysoTracker Red, the red fluorescence intensity of acidic organelles was measured. In acridine orange-stained cells, acidic lysosomal content was quantified as the mean red-to-green fluorescence ratio. Results are presented relative to untreated control cells, which were arbitrarily set to 1.

### 2.6. Caspase Activation, Mitochondrial Membrane Potential and Superoxide Measurement

Caspase activation was assessed using the cell-permeable, FITC-conjugated pan-caspase inhibitor Z-VAD-FMK (ApoStat; R&D Systems, Minneapolis, MN, USA), by quantifying the increase in green fluorescence intensity. MMP was evaluated using JC-1 dye (MedChemExpress). In polarized mitochondria, JC-1 forms red fluorescent aggregates, whereas in depolarized mitochondria, it remains in the green fluorescent monomeric form. MMP changes were expressed as the ratio of green-to-red fluorescence. Mitochondrial superoxide anion radical (O_2_•^−^) was detected using MitoSOX Red (MedChemExpress), which selectively reacts with O_2_•^−^ to emit red fluorescence proportional to its concentration. All fluorescent probes were used according to the manufacturers’ instructions for flow cytometry staining, but prior to fluorescence measurement on a Hidex Sense microplate reader (Hidex, Turku, Finland) with the instrument’s acquisition software (v1.3.0), stained cells were transferred into black 96-well plates. Fluorescence was measured using appropriate filter sets with excitation/emission wavelengths (Ex/Em) of 488/530 nm for FITC-labeled Z-VAD-FMK and JC-1 monomers, 540/590 nm for JC-1 aggregates, and 510/580 nm for MitoSOX Red. Caspase activity and MitoSOX Red fluorescence were normalized to DAPI fluorescence (Thermo Fisher Scientific; Ex/Em 358/461 nm) as an estimate of cell number per well. All results are presented as fold change relative to untreated control cells arbitrarily set to 1.

### 2.7. Measurements of Oxygen Consumption and Extracellular Acidification Rates

OXPHOS activity and glycolytic flux in A375 melanoma cells were assessed by measuring OCR and ECAR, using the MitoXpress Xtra Oxygen Consumption Assay and the pH-Xtra Glycolysis Assay, respectively (both from Agilent, Santa Clara, CA, USA), according to the manufacturer’s instructions. Fluorescence was recorded in real time for 1 h using a Hidex Sense microplate reader, starting 2 h after treatment. Time-resolved fluorescence was read with Ex/Em 380/645 nm for MitoXpress-Xtra and Ex/Em 360/620 nm for pH-Xtra. The rate of signal change (slope) in the linear region of the curve was calculated for each condition and used as a measure of OCR and ECAR. For OCR, 10 µM antimycin A (MedChemExpress) + 2 µM rotenone (Merck) was applied as a negative control, and its slope was subtracted from those of the treatment groups. For ECAR, high 250 mM 2DG was used as a negative control, and its slope was subtracted from those of the treatment groups. For comparative analysis, values were normalized to the slope obtained in the untreated control group, which was set to 1.

### 2.8. Measurement of Intracellular Calcium Levels

Intracellular calcium (Ca^2+^) levels were measured using the fluorescent probe FLUO-4 AM (Thermo Fisher Scientific) according to the manufacturer’s instructions for flow cytometry staining. Prior to fluorescence measurement on a Hidex Sense microplate reader, stained cells were transferred into black, clear-bottom 96-well plates. Fluorescence was measured at Ex/Em 485/535 nm in eight cycles of 10 min each. Relative changes in intracellular Ca^2+^ were expressed as ΔF/F_0_, where F_0_ represents baseline fluorescence before treatment. To account for differences in cell number, FLUO-4 fluorescence was normalized to DAPI fluorescence intensity, measured in parallel in the same wells.

### 2.9. Intracellular ATP Quantification

The intracellular concentration of ATP was determined using a Hidex Sense microplate reader and the Luminescent ATP Detection Assay Kit (Abcam, Cambridge, UK), according to the manufacturer’s instructions. Luminescence values were normalized to crystal violet staining performed in a parallel plate to account for differences in cell number. Results are presented as fold change relative to untreated control cells, arbitrarily set to 1.

### 2.10. RNA Interference

Small interfering RNA (siRNA) targeting human HK2 and ATG5, as well as corresponding control siRNA (all from Santa Cruz Biotechnology), was transfected into A375 cells by electroporation in the 4D-Nucleofector X Unit, using the SF Cell Line 4D-Nucleofector X Kit and DC-135 program (Lonza, Basel, Switzerland), according to the manufacturer’s instructions. The cells were rested for 24 h before treatment.

### 2.11. Immunoblotting

The expression of HK2, ATG5, cathepsins B, C, and D was evaluated by immunoblotting exactly as previously described [36], using specific primary antibodies: anti-ATG5 (#12994), anti-cathepsin B (#31718), and anti-cathepsin D (#2284) (all from Cell Signaling Technology, Beverly, MA, USA), or anti-cathepsin C (sc-74590) and anti-HK2 (sc-130358) (both from Santa Cruz Biotechnology). β-Actin (#4967S, Cell Signaling Technology) and glyceraldehyde-3-phosphate dehydrogenase (GAPDH) (MA5-15738; Thermo Fisher Scientific) were used as a loading controls. Peroxidase-conjugated anti-rabbit IgG (111-035-144) or anti-mouse IgG (115-035-146) (both from Jackson ImmunoResearch, West Grove, PA, USA) were used as secondary antibodies. Protein bands were visualized by enhanced chemiluminescence on a ChemiDoc MP Imaging System, and the signal intensity was quantified by densitometry using the Image Lab 5.0 software (both from Bio-Rad Laboratories, Hercules, CA, USA).

### 2.12. In Silico Analysis of Gene Expression

Raw gene expression data were retrieved from the GEO dataset GSE3189 (platform: Affymetrix U133A; accession GDS1375). Clinical/pathological characteristics of patients are described in the original publication by Talantov et al. [58]. Probe sets without valid gene annotation or flagged as controls were excluded. Gene annotations were taken from the corresponding GPL platform. In cases where multiple probe sets mapped to the same gene, values were collapsed to a single gene using the avereps function from the limma (v3.64.3) package. Differential expression analysis between primary melanoma and normal skin was performed in R (version 4.5.1, R Foundation for Statistical Computing, Vienna, Austria; Bioconductor (v3.21)) using the limma package (linear models with empirical Bayes). Log_2_ fold-changes (log_2_FC) and *p*-values were computed, and the false discovery rate (FDR) was adjusted by the Benjamini–Hochberg method. Volcano plots were generated in ggplot2 (v3.5.2), with log_2_FC on the x-axis and −log_10_ (*p*-value) on the y-axis. Genes with a FDR < 0.05 were considered significantly differentially expressed. The full list of analyzed genes, including log_2_FC, *p*-values, and FDR values, is provided in Appendix A.

### 2.13. Statistical Analysis

Statistical significance of differences between treatments was assessed using Student’s t-test or one-way ANOVA followed by Tukey’s post hoc test for multiple comparisons, unless otherwise stated, in GraphPad Prism. A *p* value < 0.05 was considered statistically significant.

## 3. Results

### 3.1. 2DG and LLOMe Synergistically Reduce Viability of A375 Melanoma Cells

Given the upregulated glycolysis [5,6] and expanded lysosomal compartment [19,20,21] observed in melanoma cells, we investigated whether their simultaneous targeting could synergistically reduce cell viability. The cell viability of A375 cells treated with increasing concentrations of 2DG and LLOMe for 24 and 48 h was evaluated using the crystal violet assay. The results showed that cell viability decreased in a dose- and time-dependent manner for both compounds individually, and more pronouncedly when used in combination (Figure 1A,C). To quantify potential synergistic effects, we calculated α-indices, where values above 1 indicate synergy (Figure 1B,D). Based on the α-index, the combination of 5 mM 2DG and 1 mM LLOMe was selected for subsequent mechanistic analyses. Together, these findings demonstrate that co-targeting glycolysis and lysosomal integrity synergistically reduces melanoma cell viability in a dose- and time-dependent manner, highlighting a potentially exploitable vulnerability in melanoma cells.

### 3.2. 2DG+LLOMe Induces Mixed Apoptotic and Necrotic Death in Melanoma Cells

We next investigated the mode of cell death induced by 2DG, LLOMe, and their combination. Ferrostatin-1 and necrostatin-1 failed to protect A375 cells from 2DG+LLOMe-induced cytotoxicity, suggesting that ferroptosis and necroptosis are not involved in the antimelanoma effect of the combined treatment (Figure 2A,B). Furthermore, autophagy inhibition by either wortmannin or ATG5 knockdown did not affect the viability of cells treated with 2DG, LLOMe, or their combination (Figure 2C,D), suggesting that the cytotoxic effects of these treatments are independent of autophagy modulation. Both 2DG and LLOMe induced caspase activation, with the strongest effect observed upon their combination, as demonstrated by fluorimetry in Apostat-stained cells (Figure 2F). Annexin V-FITC/7-AAD flow cytometry showed that 2DG increased the proportion of Annexin V^+^/7-AAD^−^ apoptotic cells, while LLOMe elevating both apoptotic and Annexin V^+^/7-AAD^+^ necrotic populations, with the combined treatment inducing the highest levels of apoptosis and necrosis (Figure 2E). Moreover, the two agents synergistically induced apoptosis-associated DNA fragmentation, as evidenced by an increased sub-G0/G1 population in cell cycle analysis (Figure 2G). Finally, inhibition of caspases by Q-VD-OPh reduced the cytotoxicity induced by 2DG and/or LLOMe, supporting a key role for caspase-mediated apoptosis in these treatments (Figure 2H). Collectively, our findings demonstrate that 2DG+LLOMe induce caspase-mediated apoptosis accompanied by necrosis, while ferroptosis, necroptosis, and autophagy are not involved.

### 3.3. Antimelanoma Effect of 2DG+LLOMe Is Mediated by Lysosomal Destabilization

As LLOMe is a well-known lysosome-destabilizing agent [38], we next investigated whether LMP contributes to cell death induced by the combination of 2DG and LLOMe. Phase-contrast microscopy revealed that cells treated with LLOMe or 2DG+LLOMe exhibited vacuole-like intracellular structures as early as 30 min after treatment, which may represent swollen lysosomes (Figure 3A). Fluorescence microscopy (Figure 3B) and flow cytometry (Figure 3C) showed that LLOMe alone or in combination with 2DG induced a red-to-green fluorescence shift of acridine orange 30 min after treatment, indicating decreased acidity of lysosomes or autolysosomes. Bafilomycin A1, an inhibitor of lysosomal acidification and the entry of acidophilic agents into lysosomes [59], and MG132, a proteasome inhibitor that also targets cysteine cathepsins B, C, L, and S [60,61,62,63], suppressed the cytotoxicity of LLOMe-containing treatments (Figure 3D,E). The toxicity of LLOMe and 2DG+LLOMe was not reduced by pepstatin A, an inhibitor of aspartic proteases CTSD and CTSE [64,65] (Figure 3F). On the other hand, the Ca^2+^ chelator BAPTA-AM further enhanced LLOMe- and 2DG+LLOMe-induced cytotoxicity (Figure 3G), likely due to the requirement of Ca^2+^ for lysosomal membrane repair [38,66]. Accordingly, FLUO-4 AM fluorescence measurements revealed that LLOMe, alone or in combination with 2DG, induced a time-dependent increase in intracellular Ca^2+^ levels (Figure 3H). Furthermore, LLOMe ± 2DG induced lysosomal deacidification, as demonstrated by the reduction in LysoTracker Red fluorescence, which was further enhanced in the presence of BAPTA-AM, indicating additional lysosomal damage under Ca^2+^-depleted conditions (Figure 3I). These findings indicate that the entry of LLOMe into lysosomes, lysosomal destabilization, and the activity of cysteine cathepsins are essential for the cytotoxic effect of LLOMe, both alone and in combination with 2DG.

### 3.4. 2DG+LLOMe-Induced Cell Death Is Mediated by LMP-Dependent Mitochondrial Depolarization and Oxidative Stress

Apoptosis and necrosis are frequently associated with oxidative stress [67,68] and MMP loss [69,70]. We next examined if these processes are involved in 2DG- and/or LLOMe-induced cytotoxicity. An increased green-to-red fluorescence ratio of the JC-1 dye, measured using fluorimetry, demonstrated that LLOMe, both alone and in combination with 2DG, induced significant MMP loss, which was prevented by bafilomycin A1 and MG132, inhibitors of lysosomal acidification and cysteine cathepsins, respectively (Figure 4A). Furthermore, as evidenced by enhanced fluorescence in cells stained with MitoSOX Red, LLOMe-containing treatments increased mitochondrial O_2_•^−^ production, which was also attenuated in the presence of bafilomycin A1 and MG132 (Figure 4B). Pretreatment with the antioxidant N-acetyl cysteine (NAC) partially rescued cell viability, indicating that ROS contributes to LLOMe ± 2DG-induced cytotoxicity (Figure 4C). On the other hand, pretreatment with superoxide dismutase (SOD) exacerbated LLOMe ± 2DG-induced cytotoxicity, indicating that O_2_•^−^-derived oxidants, rather than O_2_•^−^ itself, mediate LLOMe-induced cell death (Figure 4C). Together, these findings indicate that cell death induced by LLOMe, both alone and in combination with 2DG, is mediated, at least in part, by LMP, which leads to mitochondrial damage manifested by membrane depolarization and O_2_•^−^ production.

### 3.5. Combined Glycolytic and Mitochondrial Inhibition by 2DG and LLOMe Triggers Energetic Collapse and Loss of Viability

Since MMP loss is typically associated with mitochondrial dysfunction, we next measured OXPHOS following treatment with 2DG, LLOMe, or their combination. OCR measurements showed that LLOMe, both alone and more strongly in combination with 2DG, reduced OXPHOS (Appendix A and Figure 5A). In parallel, ECAR analysis revealed that 2DG strongly and LLOMe moderately suppressed glycolysis, while the most pronounced reduction in glycolytic activity was observed with the 2DG+LLOMe combination (Appendix A and Figure 5A). Accordingly, treatment with 2DG, more prominently with LLOMe, and most strongly with their combination led to a gradual, time-dependent reduction in ATP levels (Figure 5B). OXPHOS enhancers L-carnitine and succinate [71,72] partially rescued cell viability reduced by 2DG+LLOMe (Figure 5C), indicating that energy depletion contributed to the cytotoxic effect of the combined treatment. Furthermore, LLOMe synergized with other glycolytic inhibitors, pyruvate dehydrogenase kinase (PDK) inhibitor dichloroacetate (DCA) [73] and pyruvate kinase M2 inhibitor (PKM2) shikonin (SHI) [74], in exerting an antimelanoma effect (Figure 5D,E). In addition, LLOMe potentiated the cytotoxicity induced by genetic knockdown of HK2, a key glycolytic enzyme and the primary target of 2DG [75] (Figure 5F), indicating that glycolysis inhibition by 2DG is critical for the synergistic antimelanoma effect with LLOMe. These results demonstrate that 2DG+LLOMe-induced cell death is driven by severe energy depletion resulting from combined inhibition of glycolysis and OXPHOS.

### 3.6. LLOMe Exhibits Non-Selective Toxicity in Contrast to Mefloquine and Siramesine

To evaluate the therapeutic potential and selectivity of the 2DG+LLOMe combination, we compared its cytotoxicity in A375 and B16 melanoma cells to that in primary human dermal fibroblasts NHDF. As demonstrated by crystal violet test, fibroblasts were less sensitive to 2DG than A375 and B16 cells, but significantly more sensitive to LLOMe (Figure 6A). Notably, both A375 cells and fibroblasts exhibited similar sensitivity to the 2DG+LLOMe combination, whereas B16 cells were less sensitive (Figure 6A), arguing against its therapeutic applicability. Therefore, we tested the melanoma selectivity of two other LMP-inducing agents, mefloquine and siramesine. As shown in Figure 6B, fibroblasts were less sensitive to mefloquine, and 2DG+mefloquine, than A375 and B16 cells. Moreover, siramesine alone killed B16 cells more efficiently, while A375 cells and fibroblasts were affected to a similar extent (Figure 6C). However, both melanoma cell lines were more sensitive than fibroblasts to the 2DG+siramesine combination. Considering that LLOMe-induced LMP requires cathepsin C, whereas mefloquine and siramesine act independently of this protease, we hypothesized that the limited sensitivity of melanoma cells to LLOMe might be due to reduced cathepsin C expression. To test this, we first analyzed publicly available gene expression data (GSE3189) to compare expression of glycolytic enzymes and lysosomal proteases in normal skin and primary melanoma in patients. The results indeed revealed approximately a twofold lower expression of cathepsin C in malignant melanoma compared to normal skin (Figure 6D and Appendix A). In addition, expression of cathepsins E, G, K, and V was significantly reduced, whereas cathepsins A, B, D, H, and Z were upregulated in melanoma compared to normal skin. Notably, the principal executors of LMP-mediated cytotoxicity cathepsins B and D were increased by ~12-fold and ~3.5-fold, respectively (Appendix A). Analysis of glycolytic genes revealed upregulation of HK3, PFKL, ALDOA, GPI, GAPDH, PGK1, PFKL, ENO2, PKM, TPI1, PGAM1, and LDHA, with GAPDH and ENO1 showing particularly strong increases. In contrast to HK2, which was not significantly changed, the other 2-DG substrate, GPI, was increased by ~7.6-fold in melanoma (Appendix A). Moreover, ALDOC and ENO3 were significantly downregulated (Figure 6D and Appendix A). We next compared the expression of HK2 and major cathepsins between A375 cells and fibroblasts. Unlike melanoma samples from patients, where HK2 expression was not altered relative to normal skin, A375 cells exhibited higher HK2 expression compared to skin fibroblasts (Figure 6E), which may explain their increased sensitivity to 2DG. Reduced expression of cathepsin C in A375 cells was consistent with patient-derived data and could account for their lower sensitivity to LLOMe compared to fibroblasts. However, in contrast to patient samples, cathepsins B and D were also markedly less expressed in A375 cells than in fibroblasts (Figure 6E), which was in line with their increased sensitivity to LLOMe, but not with their reduced sensitivity to mefloquine and siramesine. Nevertheless, measurement of lysosomal acidity revealed that fibroblasts were less sensitive to lysosomal deacidification in the presence of mefloquine and siramesine (Figure 6F), indicating greater stability of their lysosomal membranes. These findings suggest that, unlike LLOMe, the combination of 2DG with mefloquine or siramesine exhibits greater selectivity toward melanoma cells and may represent a more promising therapeutic approach.

## 4. Discussion

This study shows that disruption of lysosomal integrity and glycolysis inhibition triggers metabolic collapse and cell death in melanoma cells. LLOMe impaired mitochondrial function through cysteine cathepsins, suppressing OXPHOS and partially inhibiting glycolysis, while 2DG blocked glycolysis. Their combination caused severe energy depletion and mixed apoptotic and necrotic death. However, the non-selective toxicity of LLOMe limits therapeutic use, underscoring the need for safer LMP inducers such as mefloquine or siramesine in combination with glycolysis inhibitors.

Consistent with our previous findings with 2DG and lysosomal detergent N-dodecylimidazole (NDI) in B16 cells [23], 2DG and LLOMe produced a synergistic antimelanoma effect against B16 and A375 melanoma cells (Figure 1A–D and Figure 6), which was not mediated by necroptosis or ferroptosis (Figure 2A,B). 2DG has been shown to stimulate [76,77] and inhibit autophagy [78]. Although lysosomal damage is generally expected to impair autophagy [79], no evidence supports such an effect for LLOMe, which instead, induces cytoprotective lysophagy [38,80,81,82,83,84]. It is therefore plausible that 2DG interferes with LLOMe-induced lysophagy. However, neither pharmacologic nor genetic inhibition of autophagy affected the cytotoxicity of 2DG, LLOMe, or their combination (Figure 2C,D), arguing against autophagy involvement. 2DG induces apoptosis [85,86], while LLOMe can trigger either apoptosis [31,38,87,88] or necrosis [33,35,36,63], depending on the dose and cell type. Consistent with this, 2DG induced caspase-dependent apoptosis, LLOMe caused both apoptosis and necrosis, and their combination triggered the strongest mixed cell death (Figure 2E–H). Given that apoptosis is a highly energy-dependent process, unlike necrosis [89], the harsh energy depletion caused by LLOMe ± 2DG likely shifted part of the apoptotic response toward necrosis.

LLOMe is an acidophilic compound that is cleaved and polymerized by cathepsin C inside lysosomes, generating membranolytic products that trigger LMP [30], which may be followed by the proteolytic activity of cysteine cathepsins B, L, S, K, and H in the cytoplasm [34] or mitochondria [32]. Accordingly, we demonstrated that LLOMe ± 2DG induced rapid lysosomal deacidification (Figure 3B,C,I). Cytotoxicity of LLOMe ± 2DG was partially prevented by the V-ATPase inhibitor bafilomycin A1 (Figure 3D), which probably prevented accumulation of acidophilic LLOMe inside lysosomes, as reported for other acidophilic drugs [59]. LLOMe-induced LMP is followed by the release of lysosomal Ca^2+^ into the cytoplasm, which activates lysosomal membrane repair machinery [38,66]. Consistently, we demonstrated that LLOMe increased intracellular Ca^2+^ levels (Figure 3H), whereas Ca^2+^ chelation further enhanced both lysosomal destabilization (Figure 3I) and cell death induced by LLOMe (Figure 3G). Partial protection was also observed with MG132 (Figure 3E), a proteasome inhibitor that also targets cathepsins B, C, L, and S [60,61,62,63], suggesting that one or more cysteine cathepsins may contribute to the observed cytotoxicity. The association of lower cathepsin B and C levels with reduced fibroblast sensitivity to LLOMe (Figure 6E) also supports their role in LLOMe-mediated cytotoxicity. Unlike a previous report implicating cathepsin D in LLOMe cytotoxicity [90], its involvement can be excluded since the cathepsin D inhibitor pepstatin A did not protect A375 cells from LLOMe ± 2DG toxicity (Figure 3F).

Mitochondrial depolarization is a key event in the intrinsic apoptotic pathway, but it can also indicate irreversible mitochondrial dysfunction leading to necrosis when ATP levels are insufficient to support apoptotic execution [69]. In accordance with previous findings [32,90], we demonstrated that LLOMe triggers MMP loss (Figure 4A). Mitochondrial depolarization was more pronounced in the presence of 2DG (Figure 4A), possibly due to the cell’s inability to maintain MMP under conditions of harsh ATP depletion [91]. The partial prevention of MMP loss by bafilomycin A1 and MG132 suggests that lysosomes and cysteine cathepsins contribute to LLOMe-induced mitochondrial depolarization. However, whether LLOMe-induced mitochondrial depolarization results from previously demonstrated cysteine cathepsin-mediated activation of pore-forming Bid and degradation of Bid inhibitors Bcl-2, Bcl-xL, and Mcl-1 [31,34,90], or from direct proteolytic damage to the outer and inner mitochondrial membrane, including components of the electron transport chain (ETC) [32], remains to be clarified.

MMP loss can enhance O_2_•^−^ production by disrupting electron flow through ETC [92], while O_2_•^−^ itself exacerbates mitochondrial dysfunction by promoting permeability transition pore opening and depolarization [93], thereby establishing a positive feedback loop. LLOMe was previously shown to stimulate both mitochondrial O_2_•^−^ production and MMP loss in a cathepsin B-dependent manner [94]. Accordingly, LLOMe, alone or in combination with 2DG, significantly increased mitochondrial O_2_•^−^ levels (Figure 4B), an effect that was partially prevented by bafilomycin A1 and MG132, further implicating lysosomal and cysteine cathepsins involvement in mitochondrial oxidative stress. Moreover, the non-specific antioxidant NAC protected against LLOMe ± 2DG-induced toxicity, while SOD exacerbated it (Figure 4C), suggesting that hydrogen peroxide (H_2_O_2_), a product of SOD-mediated O_2_•^−^ dismutation [95], or its downstream products such as highly toxic hydroxyl radical (•OH) may contribute to oxidative damage.

MMP is driving force for OXPHOS and ATP production [91]. Accordingly, LLOMe-induced MMP loss was associated with a decrease in OXPHOS activity (Appendix A and Figure 5A). Consistent with this, a previous study showed that LLOMe impaired mitochondrial respiration through the action of cysteine cathepsins [32]. While 2DG enhances OXPHOS in cells with metabolically flexible mitochondria as a compensatory response to glycolytic inhibition [96,97], it can also suppress mitochondrial respiration in highly glycolytic cells by limiting pyruvate availability for the TCA cycle [98,99]. In line with the latter, 2DG tended to reduce OXPHOS in our model, although this effect did not reach statistical significance, and the strongest OXPHOS inhibition was observed when both treatments were applied together (Appendix A and Figure 5A). Interestingly, although LLOMe was previously shown to transiently increase glycolysis in macrophages [32], it reduced glycolytic activity in our model, suggesting that its metabolic effects depends on cell type. LLOMe-mediated glycolysis inhibition could be ascribed to cathepsin-dependent degradation of glycolytic enzymes [100] and/or a global metabolic decline caused by severe cellular damage [101]. Expectedly, the glycolytic inhibitor 2DG suppressed ECAR, and the strongest inhibition was observed following the combined treatment (Appendix A and Figure 5A). L-carnitine promotes ATP production by transporting long-chain fatty acids into mitochondria for β-oxidation [102], while succinate supports mitochondrial respiration by acting as a TCA cycle intermediate and complex II electron donor [103,104]. Both energy-boosting agents partially attenuated 2DG+LLOMe-induced cytotoxicity, indicating an energy depletion-dependent cytotoxic mechanism, but the rescue was limited (Figure 5C), likely due to severe mitochondrial dysfunction. Inhibition of glycolysis by DCA, shikonin, or HK2 knockdown mimicked 2DG and enhanced LLOMe-induced cytotoxicity (Figure 5D–F), demonstrating that glycolysis interference underlies the synergy and that 2DG could be replaced by other glycolysis-targeting agents in potential therapy.

Our analysis of publicly available gene expression data is consistent with previous studies reporting upregulation of glycolytic enzymes GPI [105], ALDOA [106], GAPDH [107,108], PKM [108,109], LDHA [110,111], PGAM1 [112], and ENO1 [113], as well as downregulation of ALDOC [114], while it was first to discover upregulation of ENO2, PGK1, TPI1, PFKL, and HK3 and downregulation of ENO3 in skin melanoma patients. Furthermore, in line with previous findings [115,116,117,118,119,120,121,122], our analysis revealed upregulation of cathepsins A, B, D, and H, while cathepsin Z upregulation and downregulation of cathepsins C, E, G, K, and V in cutaneous melanoma were not previously reported (Figure 6D), indicating that these genes merit further investigation as biomarkers and therapeutic targets in cutaneous melanoma.

Although, fibroblasts represent a significant population of skin-resident cells [123], our finding that A375 cells express less cathepsin B and D but more hexokinase-2 than fibroblasts (Figure 6E) conflicts with the in silico analysis of patient samples. This discrepancy could be explained by (1) the heterogeneity of normal skin and melanoma tissues in GSE3189, which comprise multiple cell types (melanocytes/melanoma cells, fibroblasts, endothelial cells, keratinocytes, immune cells), unlike pure A375 melanoma and NHDF fibroblast cultures; (2) metabolic and lysosomal rewiring in cell lines during in vitro culture; and (3) methodological differences, as the in silico analysis assessed mRNA levels, whereas our experiments measured protein expression. However, all protein expression ratios observed in A375 vs. NHDF qualitatively align with Human Protein Atlas cell-line mRNA data, where A375 cells were compared to fibroblast models most similar to NHDF (FHDF/TERT166, BJ, BJ1-hTERT, DM-3), as direct NHDF data are not available [124].

The decreased expression of cathepsin B in A375 cells compared to fibroblasts conflicts with their increased sensitivity to toxicity of siramesine and mefloquine, reported to involve cathepsin B [43,52]. We hypothesize that the lysosomal membrane is more stable in fibroblasts than in A375 cells, as indicated by the weaker lysosomal deacidification observed upon siramesine and mefloquine treatment, which would prevent cathepsin B release and cytotoxicity. By contrast, cathepsin C acts at the very beginning of LLOMe’s cytotoxic mechanism [30]. Thus, its higher abundance in fibroblasts and the resulting increased formation of membranolytic LLOMe polymers may override the greater lysosomal membrane stability of fibroblasts, leading to the release of upregulated cathepsin B. Accordingly, monocytes were shown to be more sensitive than tumor cells to LLOMe due to their higher cathepsin C expression [31], arguing against its therapeutic applicability in cancer.

### Limitations of the Study

Most of our mechanistic experiments were conducted with LLOMe, which proved unsuitable as a potential therapeutic agent due to its non-selective toxicity. Thus, the main value of these findings lies in revealing the mechanism by which LMP and glycolysis inhibition synergize to kill melanoma cells, rather than in identifying a clinically applicable compound. By contrast, the precise mechanisms through which 2DG synergizes with siramesine and mefloquine, which appeared more promising as therapeutic options, remain to be elucidated. Furthermore, the role of individual cathepsins was not firmly established. Our data only suggest the involvement of cysteine cathepsins based on the effects of the broad-spectrum inhibitor MG132 and the correlation of cell line sensitivity with cathepsin B and C expression. It should also be noted that other inhibitors and modulators used in this study may exert additional, non-specific effects and potential drug–drug interactions (e.g., with 2DG/LLOMe), which could partly confound mechanistic interpretation. Finally, the discrepancy in gene/protein expression between patient-derived melanoma and skin samples on the one hand and fibroblast versus melanoma cell lines on the other underscores not only the limitations of our study but also the broader limitations of in vitro systems in predicting the translational relevance of anticancer therapies.

## 5. Conclusions

In summary, our results indicate that LLOMe disrupts lysosomal integrity and, through cysteine cathepsin activity, induces mitochondrial dysfunction and suppression of OXPHOS, but also inhibits glycolysis to a lesser extent through an unknown mechanism. In addition, 2DG inhibits glycolysis. The combination of 2DG and LLOMe causes a severe energy crisis, ultimately resulting in mixed apoptotic and necrotic cell death. Although LLOMe was effective, its considerable toxicity toward non-malignant cells limits its translational potential, whereas alternative lysosome-destabilizing agents such as siramesine and mefloquine show greater selectivity for melanoma cells and spare fibroblasts, while also synergizing with 2DG. These findings support the rationale for further exploration of selective LMP inducers in combination with glycolysis inhibitors as a promising therapeutic strategy against melanoma.

## Figures and Tables

**Figure 1 pharmaceutics-17-01312-f001:**
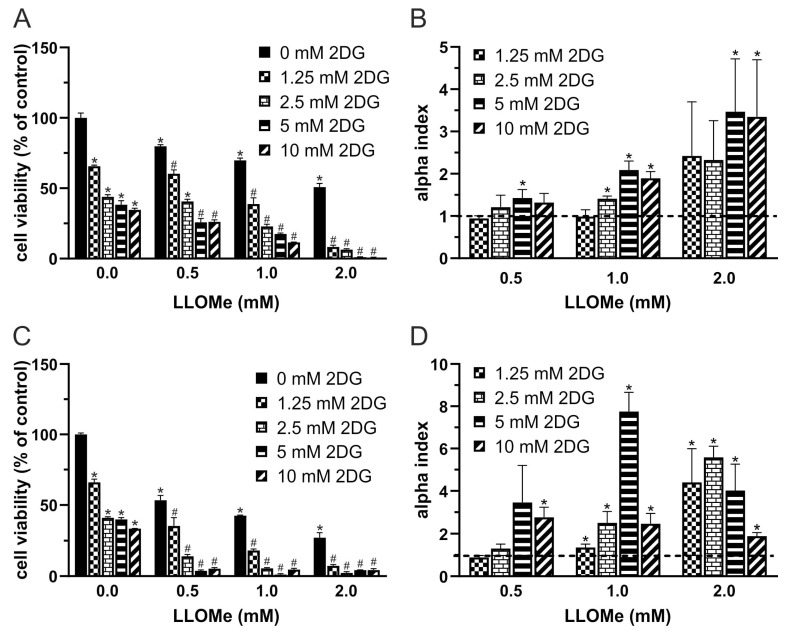
2DG and LLOMe synergistically reduce viability of A375 melanoma cells. A375 cells were treated with 2DG (1.25–10 mM) and/or LLOMe (0.5–2 mM) for 24 h (**A**,**B**) or 48 h (**C**,**D**) and cell viability was assessed using the crystal violet assay (**A**,**C**). (**A**,**C**) Data are presented as mean ± SD of triplicates from a representative experiment out of three independent repeats (* *p* < 0.05 vs. untreated control; # *p* < 0.05 vs. untreated control and single treatments with 2DG or LLOMe). (**B**,**D**) α index values were calculated from viability data in three independent experiments and are shown as mean ± SD (* *p* < 0.05 denotes α > 1, indicating synergism).

**Figure 2 pharmaceutics-17-01312-f002:**
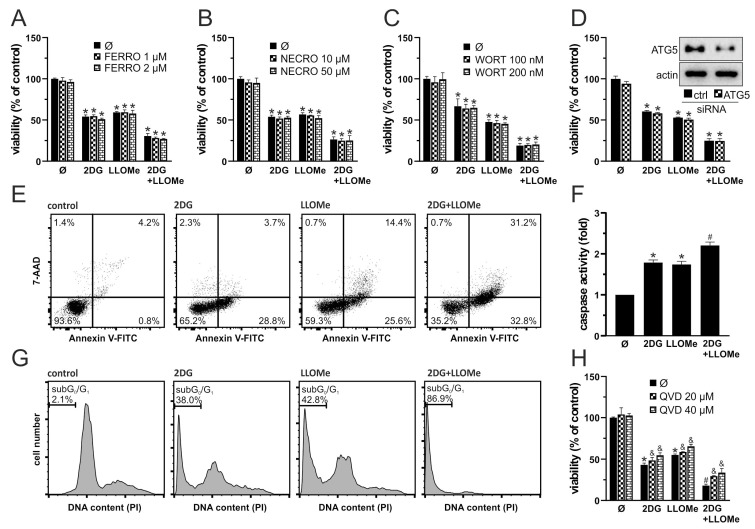
2DG+LLOMe induces mixed apoptotic and necrotic death in melanoma cells. A375 cells were treated with 5 mM 2DG and/or 1 mM LLOMe in the presence or absence of ferrostatin-1 (FERRO; (**A**)), necrostatin-1 (NECRO; (**B**)), wortmannin (WORT; (**C**)), or Q-VD-OPh (QVD; (**H**)). (**D**) A375 cells were transfected with control or ATG5-targeting siRNA prior to treatment with 5 mM 2DG and/or 1 mM LLOMe (insets show immunoblot verification of ATG5 knockdown). After 24 h cell viability was determined by crystal violet (**A**–**D**,**H**), while caspase activation was assessed by fluorimetry (**F**). Phosphatidylserine externalization (Annexin V^+^ cells), cell membrane damage (7-AAD^+^ cells) (**E**), and DNA fragmentation (sub-G_1_ compartment) (**G**) were assessed by flow cytometry after 36 h and 48 h, respectively. The representative dot plots (**E**) and histograms (**G**) are shown. The data are presented as the mean ± SD values of triplicates from a representative of three independent experiments (**A**–**D**,**F**,**H**) (* *p* < 0.05 vs. untreated control; # *p* < 0.05 vs. untreated control and single treatments with 2DG or LLOMe; and & *p* < 0.05 vs. same treatments without QVD).

**Figure 3 pharmaceutics-17-01312-f003:**
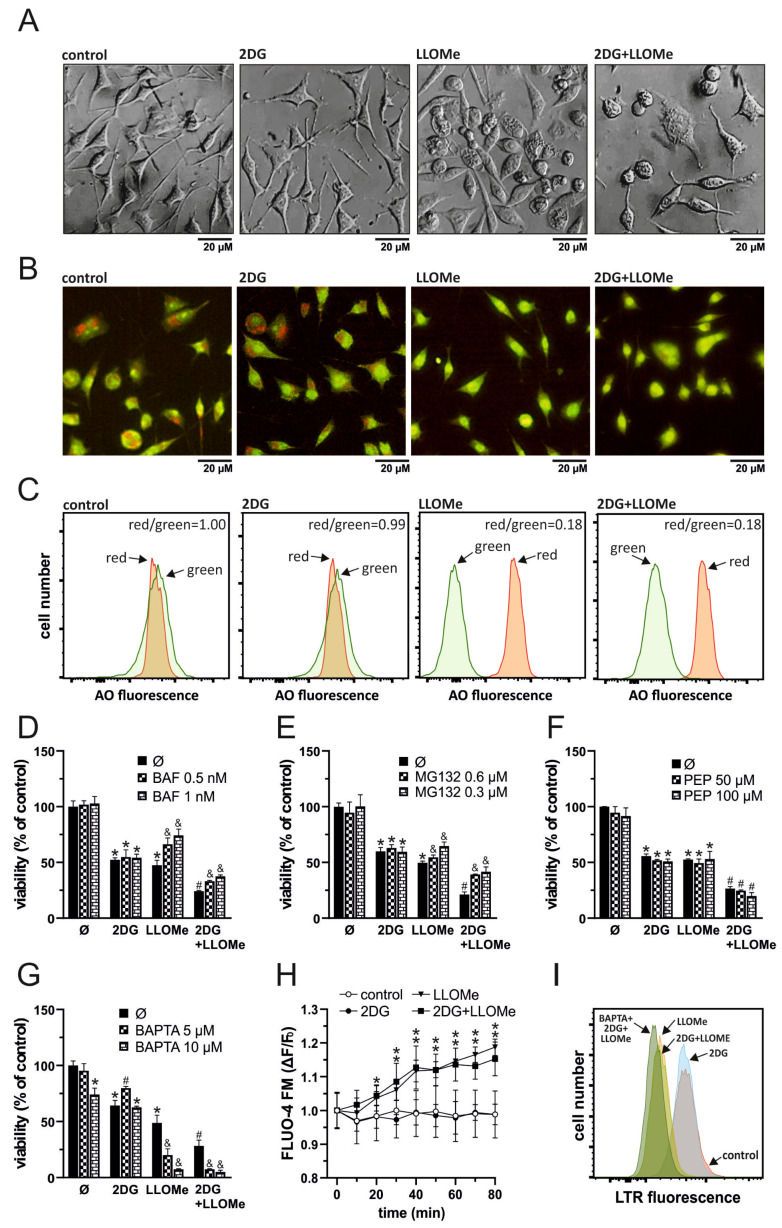
Antimelanoma effect of 2DG+LLOMe is mediated by lysosomal destabilization. A375 cells were treated with 5 mM 2DG and/or 1 mM LLOMe in the absence (**A**–**C**,**H**) or presence of bafilomycin A1 (BAF; (**D**)), MG132 (**E**), pepstatin A (PEP; (**F**)), or BAPTA-AM (BAPTA; (**G**,**I**)). After 30 min of treatment, cell morphology was examined by phase-contrast microscopy (**A**), lysosomal acidification in LysoTracker Red (LTR)-stained cells by flow cytometry (**I**), and in acridine orange (AO)-stained cells by fluorescence microscopy (**B**) or flow cytometry (**C**,**I**). After the indicated time periods, intracellular Ca^2+^ concentration was measured by fluorimetry in Fluo-4 AM-stained cells (**H**). After 24 h cell viability was determined by crystal violet (**D**–**G**). The representative micrographs (**A**,**B**) and histograms (**C**,**I**) from three independent experiments are shown. The data are presented as the mean ± SD values of triplicates from a representative of three independent experiments (**D**–**H**) (* *p* < 0.05 vs. untreated control; # *p* < 0.05 vs. untreated control and single treatments with 2DG or LLOMe; and & *p* < 0.05 vs. same treatments without BAF (**D**), MG132 (**E**), or BAPTA (**G**)).

**Figure 4 pharmaceutics-17-01312-f004:**
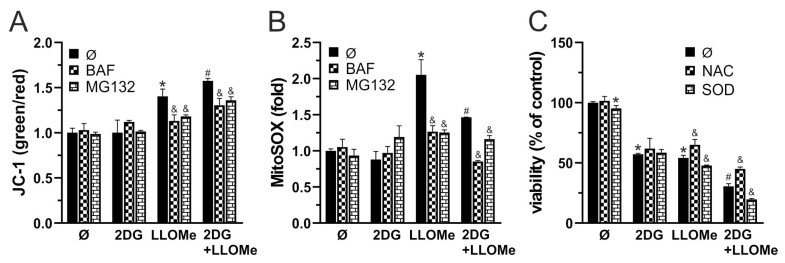
2DG+LLOMe-induced cell death is mediated by LMP-dependent mitochondrial depolarization and oxidative stress. A375 cells were treated with 5 mM 2DG and/or 1 mM LLOMe (A-C) in the presence or absence of 1 nM bafilomycin A1 (BAF) and 0.6 mM MG132 (**A**,**B**), or N-acetylcysteine (NAC; 5 mM) and superoxide dismutase (SOD; 5 µM) (**C**). After 2 h, MMP loss in JC-1-stained cells (**A**) or mitochondrial O_2_•^−^ production in MitoSOX-stained cells was assessed by fluorimetry, while after 24 h, cell viability was determined by crystal violet (**C**). The data are presented as the mean ± SD values of triplicates from a representative of three independent experiments (* *p* < 0.05 vs. untreated control; # *p* <0.05 vs. untreated control and single treatments with 2DG or LLOMe; and & *p* < 0.05 vs. same treatments without BAF and MG132 (**A**,**B**) or NAC and SOD (**C**).

**Figure 5 pharmaceutics-17-01312-f005:**
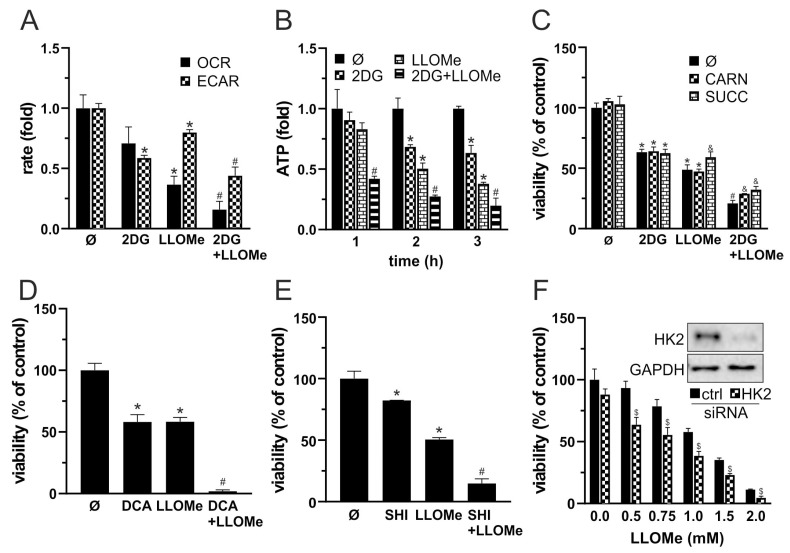
Combined glycolytic and mitochondrial inhibition by 2DG and LLOMe triggers energetic collapse and loss of viability. (**A**–**C**) A375 cells were treated with 5 mM 2DG and/or 1 mM LLOMe in the absence (**A**,**B**) or the presence of 200 µM L-carnitine (CARN) or 1 mM succinate (SUCC) (**C**). (**A**) Oxygen consumption rate (OCR) and extracellular acidification rate (ECAR) were measured using fluorescence-based assays between 2 and 3 h post treatment. (**B**) Intracellular ATP levels were measured at the indicated time points using a bioluminescence assay. (**D**,**E**) A375 cells were treated with 1 mM of LLOMe in the presence or absence of 40 mM dichloroacetate (DCA; (**D**)) or 1 mM shikonin (SHI; (**E**)). (**F**) A375 cells transfected with control or HK2-targeting siRNA were treated with 1 mM LLOMe (insets show immunoblot verification of HK2 knockdown). (**C**–**F**) After 24 h, cell viability was determined by crystal violet. (**A**–**F**) The data are presented as the mean ± SD values of triplicates from a representative of three independent experiments (* *p* < 0.05 vs. untreated control; # *p* < 0.05 vs. untreated control, and single treatments with 2DG, SHI, DCA, or LLOMe; and & *p* < 0.05 vs. same treatments without CARN or SUCC; $ *p* < 0.05 vs. control siRNA).

**Figure 6 pharmaceutics-17-01312-f006:**
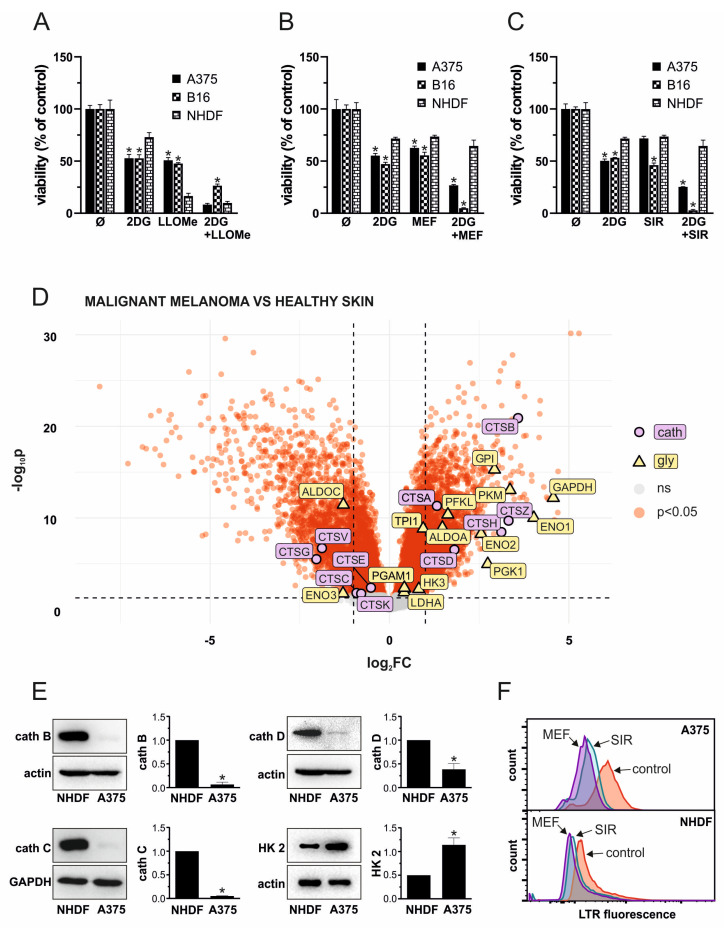
LLOMe exhibits non-selective toxicity in contrast to mefloquine and siramesine. (**A**–**C**) A375, B16, and NHDF cells were treated with 5 mM 2DG in the presence or absence of 1 mM LLOMe (**A**), 20 µM mefloquine (MEF; (**B**)), or 10 µM siramesine (SIR; (**C**)) and after 24 h, cell viability was determined by crystal violet. The data are presented as the mean ± SD values of triplicates from a representative of three independent experiments (* *p* < 0.05 vs. NHDF under the same treatment). (**D**) Data from the publicly available GEO microarray dataset GSE3189 were analyzed to compare the expression of glycolytic enzymes and lysosomal cathepsins between primary melanoma from patients and normal skin from control subjects. The volcano plot was generated in R, with genes showing FDR < 0.05 considered significantly differentially expressed (red), while non-significant genes are shown in gray. Significantly altered glycolytic enzymes are highlighted in yellow, and cathepsins in purple. The full list of analyzed genes is provided in Appendix A. (**E**) The expression of cathepsin (cath) (**B**–**D**), as well as hexokinase-2 (HK2), was evaluated in untreated NHDF and A375 cells by immunoblot. Representative immunoblots were presented and densitometry data as the mean ± SD values from three independent experiments (* *p* < 0.05 vs. NHDF). (**F**) Lysosomal deacidification in LysoTracker Red-stained NHDF and A375 cells upon 30 min treatment with 20 µM mefloquine (MEF) or 10 µM siramesine (SIR) was evaluated by flow cytometry and representative histograms from three independent experiments presented.

## Data Availability

Data presented in this study is contained within the article and Appendix A. Further inquiries can be directed at the corresponding author.

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
