# Peer review of "Targeting Glycolysis with 2-Deoxy-D-Glucose and Lysosomal Integrity with L-Leucyl-L-Leucine Methyl Ester as Antimelanoma Strategy"

_pharmaceutics, 2025, doi:10.3390/pharmaceutics17101312_

Round 1

Reviewer 1 Report

Comments and Suggestions for Authors

The authors provided a manuscript with a solid background and introduction to the topic. However, there are several areas that need clarification and improvement to strengthen the overall impact of the work. Major revisions are required in the topics below.

Abstract: The abstract currently lacks clarity due to the extensive use of forward slashes, which makes the text difficult to follow. Consider rewriting for better flow. The information provided about inhibitors is vague, which contributes to the abstract being unclear. Please revise and include specific details to make the abstract more informative and concise.

Introduction: It would be valuable to include more extensive experimental evidence supporting the specific anti-tumor effects, as well as mechanisms, of the suggested drugs, particularly in vivo data, if available.

Figures and Results

- Figure 2E: It is surprising that the control group had a higher number of necrotic cells (Annexin V⁺/PI⁺) than the separate treatment groups. Could the authors explain this unexpected result?

- Figure 3: The use of inhibitors to explore specific effects is an interesting mechanistic approach. However, the study does not address the potential effects of combining the different drugs and inhibitors on the cells. This is a limitation that should be addressed in the discussion. Additionally, could the authors provide evidence of the involvement of Ca²⁺ in the process? There are multiple available tools for assessing Ca²⁺transport in cells.

- Figure 5:Could the authors include Seahorse analysis to offer real-time insights into the effects on cellular metabolism? Furthermore, the claim that carnitine and succinate rescue cell viability is not clearly supported by the displayed results. Please clarify or provide additional supporting data.

- Figure 6: Including data from additional cancer cell lines would help validate the broader relevance of the findings.

- Figure 7: The statement regarding "global upregulation" (line 420) is not strongly supported by the data. Were glycolysis- and cathepsin-related processes identified as significant pathways in the gene ontology analysis? The current representation of gene expression is not very clear; a heatmap or volcano plot would better illustrate the differences and enrich the analysis. Finally, what would be the effect of downregulation or overexpression of CTSB and CTSD in melanoma cells, with use of siRNA for instance? Are CTSB and CTSD expression levels higher in melanoma cells compared to fibroblasts?

Discussion: The discussion section is too long and should be more concise, with a clearer focus on the key findings and their implications in relation to known literature. Also, limitations of the research should be made explicit.

Comments on the Quality of English Language

Good quality of English language.

Author Response

RESPONSE TO REVIEWER 1

The authors provided a manuscript with a solid background and introduction to the topic. However, there are several areas that need clarification and improvement to strengthen the overall impact of the work. Major revisions are required in the topics below.

Abstract: The abstract currently lacks clarity due to the extensive use of forward slashes, which makes the text difficult to follow. Consider rewriting for better flow. The information provided about inhibitors is vague, which contributes to the abstract being unclear. Please revise and include specific details to make the abstract more informative and concise.

We thank the reviewer for this helpful comment. In the revised version, we removed the forward slashes and rephrased the Abstract for improved flow and clarity. Regarding inhibitors, the Abstract is limited to 250 words, so we described them by process and functional class rather than listing individual compounds. We also added new results to Abstract.

Introduction: It would be valuable to include more extensive experimental evidence supporting the specific anti-tumor effects, as well as mechanisms, of the suggested drugs, particularly in vivo data, if available.

In the revised Introduction, we expanded the background to include additional references on the anti-tumor effects and mechanisms of 2DG (lines 61-71), LLOMe (lines 85-93), mefloquine (lines 94-101), and siramesine (lines 101-106), covering both in vitro and in vivo studies.

Figures and Results

- Figure 2E: It is surprising that the control group had a higher number of necrotic cells (Annexin V⁺/PI⁺) than the separate treatment groups. Could the authors explain this unexpected result?

We thank the reviewer for pointing out this issue. In the experiment shown in Figure 2E, the control group indeed displayed an unexpectedly higher proportion of necrotic cells, which we attribute to a technical problem affecting the control sample in that particular run. In the revised version of the manuscript, we have replaced dot plots with dot plots from other experiment.

- Figure 3: The use of inhibitors to explore specific effects is an interesting mechanistic approach. However, the study does not address the potential effects of combining the different drugs and inhibitors on the cells. This is a limitation that should be addressed in the discussion.

We agree with the reviewer that the use of different inhibitors and modulators may exert additional, non-specific effects. This point has been included as part of the “Limitation of the study” section (lines 632-637), where we noted that results obtained with various inhibitors, antioxidants, Ca²⁺ chelators, and energy-boosting agents should be interpreted with caution.

Additionally, could the authors provide evidence of the involvement of Ca²⁺ in the process? There are multiple available tools for assessing Ca²⁺transport in cells.

To address the reviewer’s comment, we performed additional fluorometry experiments using the Fluo-4 probe to detect intracellular Ca²⁺. Our results showed that LLOMe±2DG treatment increased Fluo-4 fluorescence (Fig. 3H), consistent with Ca²⁺ leakage from damaged lysosomes, as previously reported (10.1038/s41419-020-2527-8; 10.1073/pnas.2205590119). Furthermore, the Ca²⁺ chelator BAPTA-AM additionally reduced fluorescence of cells stained with acidophilic dye LysoTracker Red in LLOMe+2DG-treated cells (Fig. 3I), indicating enhanced LMP. BAPTA-A also potentiated LLOMe±2DG-induced cytotoxicity (Fig. 3G). These findings support the pivotal role of Ca²⁺ in lysosomal membrane repair, in agreement with published reports (10.1038/s41419-020-2527-8; 10.1073/pnas.2205590119). The corresponding text has been added to the revised Results (lines 340-350) and Discussion (lines 525-529).

- Figure 5:Could the authors include Seahorse analysis to offer real-time insights into the effects on cellular metabolism?

We have included the metabolic flux analysis in the Supplementary material (Supplementary Figure S1A, B). As detailed in the Methods, mitochondrial respiration and glycolytic flux were quantified in real time using the Agilent MitoXpress Xtra (OCR) and pH-Xtra (ECAR) assays on a Hidex microplate reader. Kinetic traces were analyzed within the linear segment to obtain slopes. Non-mitochondrial OCR (rotenone + antimycin A) and non-glycolytic acidification (2-deoxy-D-glucose, 250 mM) were subtracted to yield ΔOCR and ΔECAR, which were normalized to control. We have emphasized the procedures and OCR/ECAR calculations in more detail in the Methods section (lines 200-213) and in the legend of the Supplementary Figure S1.

Furthermore, the claim that carnitine and succinate rescue cell viability is not clearly supported by the displayed results. Please clarify or provide additional supporting data.

We agree with the reviewer that the protective effects of carnitine and succinate in cells treated with 2DG+LLOMe are modest. However, these effects were statistically significant (two-tailed type 2 t-test: p = 0.047 for carnitine, p = 0.049 for succinate). Both agents are known to enhance mitochondrial ATP synthesis (10.3390/nu3080735; 10.1124/jpet.107.130872; 10.1016/j.bbabio.2016.03.012), and our data indicate that LLOMe impairs mitochondrial function (Fig. 4A, B; Fig. 5A; Fig. S1A), which likely explains the limited extent of protection. This point has also been discussed in the revised manuscript (lines 578-583).

- Figure 6: Including data from additional cancer cell lines would help validate the broader relevance of the findings.

We agree with the reviewer and have now included data from an additional melanoma cell line. Specifically, the mouse melanoma B16 cells were added to the viability assays with 2DG±LLOMe, 2DG±mefloquine, and 2DG±siramesine, and plotted alongside A375 melanoma cells and NHDF fibroblasts in the same graphs (Fig. 6A-C). The B16 responses displayed the same qualitative trends as A375, thereby supporting the broader relevance of our findings at the level of cell viability. These results are now described in the Results section (lines 433-444).

- Figure 7: The statement regarding "global upregulation" (line 420) is not strongly supported by the data. Were glycolysis- and cathepsin-related processes identified as significant pathways in the gene ontology analysis?

We applied a complementary competitive gene set test (camera, limma), which confirmed significant enrichment of glycolysis genes in malignant melanoma compared to controls (FDR = 0.0009), while cathepsin genes showed a non-significant trend (FDR = 0.1213). However, GO enrichment (clusterProfiler) and KEGG pathway analysis did not identify glycolysis- or cathepsin-related processes as significantly enriched (FDR > 0.05). We agree with the reviewer and have therefore removed the statement claiming a “global upregulation of glycolytic genes and cathepsins”.

The current representation of gene expression is not very clear; a heatmap or volcano plot would better illustrate the differences and enrich the analysis.

In the revised manuscript, we have included volcano plots for the malignant vs. healthy skin comparison, with glycolytic and cathepsin genes highlighted (Fig. 6D), and provided a table with log2FC and FDR values (Supplementary Table 1). In the initial version, we also presented data from patients with benign tumors (former Fig. 7A and B). However, because patient-derived datasets overall did not consistently replicate the trends observed in cell lines, we chose to keep only the revised figures on the malignant vs. healthy comparison. This avoids diluting it with inconsistent patterns. The corresponding description is now provided in the Results section (lines 447-461) and further addressed in the Discussion (lines 588-597).

Finally, what would be the effect of downregulation or overexpression of CTSB and CTSD in melanoma cells, with use of siRNA for instance?

Genetic perturbations (e.g., siRNA knockdown or cDNA overexpression of cathepsin B and cathepsin D) would be informative but were not feasible within the revision timeframe. Instead, we assessed cathepsin function pharmacologically. Pepstatin A, a cathepsin D inhibitor, did not alter viability (Fig. 3F), arguing against a major contribution of cathepsin D to the acute cytotoxic response under our conditions. MG132 increased viability in LLOMe±2DG-treated cells (Fig. 4E). However, MG132 inhibits several cysteine cathepsins, including cathepsins B, C, and L, which are considered the main mediators of LLOMe toxicity (10.1073/pnas.87.1.83; 10.1111/febs.15326; 10.1038/s41467-022-34632-8), as well as the proteasome. Therefore, this rescue cannot be attributed with certainty to any of these cathepsins. Accordingly, we cannot conclusively predict the effects of cathepsin B or cathepsin D knockdown/overexpression, and we note this in the Limitation of the study section (lines 631-634).

Are CTSB and CTSD expression levels higher in melanoma cells compared to fibroblasts?

Consistent with our in silico analysis of melanoma tissues, immunoblotting showed that cathepsin C is reduced in melanoma cells relative to primary fibroblasts. By contrast, although cathepsins B and D are increased in melanoma tissues, they were lower in melanoma cells than in fibroblasts (Fig. 6D, E), as described in the Results (lines 466-469). Possible explanations for this discrepancy, is provided in the Discussion (lines 598-606).

Discussion: The discussion section is too long and should be more concise, with a clearer focus on the key findings and their implications in relation to known literature.

We thank the reviewer for this constructive suggestion. In the revised manuscript, the Discussion has been streamlined to sharpen the focus on the key findings and their implications within the context of the existing literature. The section was reduced from 1,692 to 1,466 words (~13% reduction) by removing redundancies, consolidating overlapping points, and relocating methodological details to the appropriate sections.

Also, limitations of the research should be made explicit.

We agree and have made the limitations explicit. In the revised manuscript we added a dedicated subsection “4.1. Limitations of the study” (lines 624-641) that: 1) notes the non-selective toxicity of LLOMe and that our primary contribution is mechanistic; 2) states that the synergy mechanisms of 2DG with siramesine/mefloquine remain to be clarified; 3) acknowledges that the roles of individual cathepsins were not firmly established; 4) cautions that pharmacological tools (e.g., inhibitors, antioxidants, Ca²⁺ chelators, energy-boosting agents) may have off-target effects; and 5) highlights discrepancies between patient tissues and cell lines, underscoring general limits of in-vitro models.

Comments on the Quality of English Language

Good quality of English language.

Submission Date

06 July 2025

Date of this review

23 Jul 2025 16:28:48

Reviewer 2 Report

Comments and Suggestions for Authors

Kosic et al aim to investigate the combinatorial effect of glycolytic inhibitor 2-deoxy-D-glucose (2DG) and cathepsin C-dependent LMP inducer L-leucyl-L-leucine methyl ester (LLOMe) on melanoma cells A375 and fibroblasts. To this extent, the authors treat both cells with varying concentrations of both agents for different time points and evaluate cell viability, cell death, mitochondrial membrane potential, oxygen consumption, and intracellular ATP levels. The authors use inhibitors to uncover the types of cell death. Additionally, the authors take advantage of the existing publicly available data sets for in silico analysis of gene expression. Although the combinatorial use of the agents appears to be more potent, the authors report that this strategy does not have any therapeutic potential since the fibroblast cells appear to be more sensitive to LLOMe than the melanoma cells. Alternatively, the authors propose the combination of 2DG with siramesine and mefloquine, as the cancer cells display higher sensitivity compared to the fibroblast cells.

Melanoma is a highly aggressive type of skin cancer with an increasing incidence. Although patients with BRAF mutations initially respond to BRAF inhibitors, relapses require the development of novel therapeutic agents. Kosic et al examine the therapeutic potency of the combinatorial use of 2DG and LLOMe. Although the authors report no such potential, the combinatorial use of 2DG with siramesine and mefloquine holds potential. I recommend the following remarks for consideration:

  1. Lines 232-235: Figure 1C: I cannot confirm the phenotypes from what is presented in Figure 1C. However, these phenotypes are more prominent in Figure 3A. Thus, either the authors should present better micrographs or simply remove Fig. 1C.
  2. I think that the transition of the combinatorial use of 2DG and LLOMe to the combinatorial use of 2DG and siramesine and mefloquine is rather sharp. Thus, it would be nice to introduce these two agents in the Introduction in an appropriate place.
  3. Lines 288-289: Fig. 3B and Fig. 3C are misused. Fig. 3B in the text should actually be Fig. 3C and vice versa.
  4. Figure 4A: Please replace FL1/FL2 on the Y axis with the fluorescence color used (Green/Red??)
  5. Line 353: Fig 5C should be Fig. 5B and line 360: Fig 5E should be Fig. 5F.
  6. It would be nice to provide the quantitative differences reported among various treatments, instead of describing the effect as “marked”, “significantly increased” “more”..etc. For example, line 404 “we observed significantly increased expression of HK…”. The increase may be statistically significant (e.g. p< 0,00001) but how many folds? 1.1-fold or 10-fold? There are many such cases in the text.

Author Response

RESPONSE TO REVIEWER 2

Kosic et al aim to investigate the combinatorial effect of glycolytic inhibitor 2-deoxy-D-glucose (2DG) and cathepsin C-dependent LMP inducer L-leucyl-L-leucine methyl ester (LLOMe) on melanoma cells A375 and fibroblasts. To this extent, the authors treat both cells with varying concentrations of both agents for different time points and evaluate cell viability, cell death, mitochondrial membrane potential, oxygen consumption, and intracellular ATP levels. The authors use inhibitors to uncover the types of cell death. Additionally, the authors take advantage of the existing publicly available data sets for in silico analysis of gene expression. Although the combinatorial use of the agents appears to be more potent, the authors report that this strategy does not have any therapeutic potential since the fibroblast cells appear to be more sensitive to LLOMe than the melanoma cells. Alternatively, the authors propose the combination of 2DG with siramesine and mefloquine, as the cancer cells display higher sensitivity compared to the fibroblast cells.

Melanoma is a highly aggressive type of skin cancer with an increasing incidence. Although patients with BRAF mutations initially respond to BRAF inhibitors, relapses require the development of novel therapeutic agents. Kosic et al examine the therapeutic potency of the combinatorial use of 2DG and LLOMe. Although the authors report no such potential, the combinatorial use of 2DG with siramesine and mefloquine holds potential. I recommend the following remarks for consideration:

  1. Lines 232-235: Figure 1C: I cannot confirm the phenotypes from what is presented in Figure 1C. However, these phenotypes are more prominent in Figure 3A. Thus, either the authors should present better micrographs or simply remove Fig. 1C.

We thank the reviewer for this comment. We agree that the phenotypes are more clearly visible in Figure 3A, and therefore we have removed Figure 1C from the revised manuscript to avoid redundancy.

  1. I think that the transition of the combinatorial use of 2DG and LLOMe to the combinatorial use of 2DG and siramesine and mefloquine is rather sharp. Thus, it would be nice to introduce these two agents in the Introduction in an appropriate place.

Per the reviewer’s suggestion, we added a dedicated paragraph to the Introduction that briefly outlines the anticancer mechanisms of mefloquine and siramesine in vitro and in vivo (lines 94-106).

  1. Lines 288-289: Fig. 3B and Fig. 3C are misused. Fig. 3B in the text should actually be Fig. 3C and vice versa.

Thank you for noting this. We have corrected the mistake.

  1. Figure 4A: Please replace FL1/FL2 on the Y axis with the fluorescence color used (Green/Red??)

We have corrected the mistake.

  1. Line 353: Fig 5C should be Fig. 5B and line 360: Fig 5E should be Fig. 5F.

We have corrected the mistake.

  1. It would be nice to provide the quantitative differences reported among various treatments, instead of describing the effect as “marked”, “significantly increased” “more”..etc. For example, line 404 “we observed significantly increased expression of HK…”. The increase may be statistically significant (e.g. p< 0,00001) but how many folds? 1.1-fold or 10-fold? There are many such cases in the text.

In accordance with Reviewer #1’s suggestion, we replaced the panel comparing malignant vs. control samples with a volcano plot. Because gene expression varies widely, the x-axis is shown as log2 fold change (log2FC), which makes exact linear fold changes less apparent. To aid interpretation, we added Supplementary Table S1 reporting linear fold changes for all mentioned genes. In the Results, we now also report fold changes for statistically significant genes of interest (cathepsins B, C, and D, and phosphoglucoisomerase) (lines 450-451, 454-455, 458-459). For other results plotted with a linear y-axis, we did not restate fold changes in the text, as they are directly readable from the figure and we wished to avoid clutter.

Round 2

Reviewer 2 Report

Comments and Suggestions for Authors

The authors have properly addressed all the points raised in the previous round of review.